# AIM2 Inflammasome in Tumor Cells as a Biomarker for Predicting the Treatment Response to Antiangiogenic Therapy in Epithelial Ovarian Cancer Patients

**DOI:** 10.3390/jcm10194529

**Published:** 2021-09-30

**Authors:** Po-Chao Hsu, Tai-Kuang Chao, Yu-Ching Chou, Mu-Hsien Yu, Yu-Chi Wang, Yi-Hsin Lin, Yi-Liang Lee, Li-Chun Liu, Cheng-Chang Chang

**Affiliations:** 1Department of Obstetrics and Gynecology, Tri-Service General Hospital, National Defense Medical Center, Taipei 114, Taiwan; raynhsu@gmail.com (P.-C.H.); hsienhui@ms15.hinet.net (M.-H.Y.); yuchitsgh@gmail.com (Y.-C.W.); m860371@gmail.com (Y.-H.L.); lylobgyn@gmail.com (Y.-L.L.); lvita.tw@gmail.com (L.-C.L.); 2Division of Obstetrics and Gynecology, Tri-Service General Hospital, Penghu Branch, Magong City 880, Taiwan; 3Department of Pathology, Tri-Service General Hospital, National Defense Medical Center, Taipei 114, Taiwan; chaotai.kuang@msa.hinet.net; 4School of Public Health, National Defense Medical Center, Taipei 114, Taiwan; trishow@mail.ndmctsgh.edu.tw; 5Division of Obstetrics and Gynecology, Tri-Service General Hospital Songshan Branch, Taipei 105, Taiwan

**Keywords:** epithelial ovarian cancer (EOC), antiangiogenic therapy, bevacizumab, complement C3, C5, AIM2 inflammasome

## Abstract

Antiangiogenic therapy, such as bevacizumab (BEV), has improved progression-free survival (PFS) and overall survival (OS) in high-risk patients with epithelial ovarian cancer (EOC) according to several clinical trials. Clinically, no reliable molecular biomarker is available to predict the treatment response to antiangiogenic therapy. Immune-related proteins can indirectly contribute to angiogenesis by regulating stromal cells in the tumor microenvironment. This study was performed to search biomarkers for prediction of the BEV treatment response in EOC patients. We conducted a hospital-based retrospective study from March 2013 to May 2020. Tissues from 78 Taiwanese patients who were newly diagnosed with EOC and peritoneal serous papillary carcinoma (PSPC) and received BEV therapy were collected. We used immunohistochemistry (IHC) staining and analyzed the expression of these putative biomarkers (complement component 3 (C3), complement component 5 (C5), and absent in melanoma 2 (AIM2)) based on the staining area and intensity of the color reaction to predict BEV efficacy in EOC patients. The immunostaining scores of AIM2 were significantly higher in the BEV-resistant group (RG) than in the BEV-sensitive group (SG) (355.5 vs. 297.1, *p* < 0.001). A high level of AIM2 (mean value > 310) conferred worse PFS after treatment with BEV than a low level of AIM2 (13.58 vs. 19.36 months, adjusted hazard ratio (HR) = 4.44, 95% confidence interval (CI) = 2.01–9.80, *p* < 0.001). There were no significant differences in C3 (*p* = 0.077) or C5 (*p* = 0.326) regarding BEV efficacy. AIM2 inflammasome expression can be a histopathological biomarker to predict the antiangiogenic therapy benefit in EOC patients. The molecular mechanism requires further investigation.

## 1. Introduction

In 2018, approximately 295,000 females were diagnosed with ovarian cancer, and almost 185,000 worldwide died from this disease [1]. Approximately 75% of women have advanced-stage epithelial ovarian cancer (EOC) at diagnosis due to asymptomatic features at an early stage [2]. EOC constitutes the seventh most commonly diagnosed cancer among women worldwide, with a 46% 5-year survival rate after diagnosis [2,3]. Primary cytoreduction surgery followed by systemic chemotherapy is the standard initial therapy for women with advanced-stage EOC.

Angiogenesis is a complex process under the regulation of multiple signaling pathways [4]. There is widespread knowledge that vascular endothelial growth factor (VEGF) is the most reliable marker of angiogenesis [5]. There is strong evidence that VEGF stimulates tumor growth, ascites, and metastases. Therefore, VEGF inhibition has become a therapeutic target in patients with EOC; for example, the VEGF inhibitor bevacizumab (BEV) has been evaluated in the treatment of EOC in several randomized phase III trials [6,7,8,9,10]. The GOG-218 trial revealed a progression-free survival (PFS) benefit only in women who received concurrent and maintenance BEV compared with those who received chemotherapy alone, and there was no difference in overall survival (OS). The improvement in PFS was 3.8 months (median PFS: 10.3 months for chemotherapy, 14.1 months for maintenance therapy, *p* < 0.001) [10]. ICON7 also evaluated BEV incorporation into first-line therapy with a 2-arm trial of carboplatin/paclitaxel (CP)/BEV and BEV maintenance vs. CP/placebo and placebo maintenance. In high-risk patients, PFS and OS were enhanced in the BEV arm versus the chemotherapy alone arm (restricted mean PFS: 20.0 months vs. 15.9 months, log-rank *p* = 0.001; restricted mean OS: 39.3 months vs. 34.5 months, log-rank *p* = 0.03) [6]. The GOG-213 study demonstrated that in recurrent EOC, the addition of BEV to chemotherapy resulted in an OS benefit of five months compared to chemotherapy alone (median OS: 42.2 months vs. 37.3 months, *p* = 0.056). The GOG-213 study showed that women lived a median 3.4 months longer without disease progression with the addition of BEV to chemotherapy compared to chemotherapy alone (median PFS: 13.8 months vs. 10.4 months, *p* < 0.0001) [9].

A study of 61 patients with EOC or primary peritoneal cancer treated with BEV showed that high baseline plasma VEGF levels were associated with short median survival and an increased risk of death [11,12]. VEGF-A plays a dominant role in angiogenesis among the VEGF family of ligands [12]. A study indicated that plasma VEGF-A levels could predict patient outcome but not the effect of BEV in patients with colorectal cancer (CRC), non-small cell lung cancer, and renal cell carcinoma [13]. VEGFR-2 is the most important receptor for VEGF-A-mediated angiogenesis [14]. Nevertheless, plasma VEGFR-2 was not predictive of clinical outcomes for women with EOC who received BEV treatment in the GOG-218 trial [12].

Absent in melanoma 2 (AIM2), a cytosolic dsDNA sensor, can induce the inflammasome upon the intracellular delivery of double-stranded DNA (dsDNA) to protect cells from pathogenic attack, including those from bacteria, viruses, fungi, and parasites [15]. By recognizing dsDNA, AIM2 promotes the assembly of a multiprotein oligomeric complex called the inflammasome [16]. Aberrant inflammasome signaling is associated with chronic inflammatory and metabolic diseases, neurodegeneration, and cancer [16,17]. Several previous studies revealed that AIM2 inflammasome can suppress colorectal tumorigenesis, hepatocellular carcinoma and breast cancer [16,17]. On the other hand, knockdown of AIM2 inflammasome resulted in the suppression of growth and vascularization of cutaneous squamous cell carcinoma xenografts in vivo [18]. In our previous study, initiating proteins of the AIM2 inflammasome were significantly correlated with an inferior prognosis and contributed to EOC progression. We established a pathogenic pathway and demonstrated a critical role of the dysregulated inflammasome in modulating the malignant transformation of EOC [15,19]. Complement systems are thought to play a critical role in the development of several cancers related to chronic inflammation, such as lung, liver, gastrointestinal, cervical, and ovarian cancers [20]. In ascites of ovarian cancer patients, high complement anaphylatoxin levels suggestive of local complement activation have been observed [21]. Immune-related genes involved in the complement system had dual effects on patient survival, and immunohistochemical examinations revealed high expression levels of complement component 3a receptor (C3aR) and complement component 5a receptor (C5aR) in clear cell carcinoma of the ovary [22]. However, the roles of the inflammasome and complement system in the anti-VEGF treatment of ovarian cancer have yet to be determined.

Current evidence supports the use of BEV in ovarian cancer that patients do not have complete cytoreduction surgery at stage III; patients at stage IV or recurrence where BRCA mutations are not confirmed [6,10]. However, there are no available biomarkers for the prediction of BEV treatment response for routine clinical use. This study will validate the role of immune-related proteins in the treatment response to antiangiogenic agents in EOC patients.

## 2. Materials and Methods

### 2.1. Patients and Specimens

From March 2013 to May 2020, 78 patients with histologically proven EOC and peritoneal serous papillary carcinoma (PSPC) and with paraffin-embedded tissue sections of the primary ovarian tumors and omental and peritoneal metastases were examined. Total 431 sections from the tumor samples were evaluated by IHC to assess expression patterns of AIM2, complement component 3 (C3), and complement component 5 (C5) as well as the associations between these molecular biomarkers with patient clinicopathological parameters and survival following BEV therapy (Table 1). These patients had complete medical records and had received debulking surgery and chemotherapy with BEV treatment at Tri-Service General Hospital. Two groups of patients were compared: the BEV-sensitive group (SG) and the BEV-resistant group (RG). The study flow chart is shown in Figure 1. These 78 patients had an Eastern Cooperative Oncology Group (ECOG) performance status of 0 or 1 before BEV therapy. The patients with ovarian cancer stages I and II (Total *n* = 12) all received complete cytoreduction surgery (completeness of cytoreduction score-CC 0/1) at the beginning for primary cancer but experienced recurrence. After recurrence, 2 patients received cytoreductive surgery and hyperthermic intraperitoneal chemotherapy (CRS + HIPEC) followed by CP + BEV and the other 10 patients received CP + BEV without further surgery. Patients with stage III ovarian cancer (*n* = 50) received upfront debulking surgery at first. 28 patients received the complete cytoreduction in which 16 patients underwent chemotherapy with CP and 12 patients received CP + BEV as adjuvant treatment after operation. Because of severe pelvic adhesion, severe intestinal adhesion, metastatic lymph nodes with aortic invasion and severe peritoneal carcinomatosis, other 22 patients received the suboptimal debulking surgery and then CP with or without BEV. The subgroup of primary stage III ovarian cancer (*n* = 29) received the concurrent BEV therapy with or without maintenance BEV. Patients with stage III ovarian cancer in the subgroup of recurrence (*n* = 21) received CRS + HIPEC followed by CP + BEV (*n* = 15) or CP + BEV without further surgery (*n* = 6). The patients with stages IV ovarian cancer (*n* = 16) received upfront debulking surgery (complete cytoreduction, *n* = 2; suboptimal, *n* = 14). The subgroup of primary stage IV ovarian cancer (*n* = 7) received the concurrent BEV therapy with or without maintenance BEV. Patients with stage IV ovarian cancer in the subgroup of recurrence (*n* = 9) received CRS + HIPEC followed by CP + BEV (*n* = 6) or CP + BEV without further surgery (*n* = 3). Total 36 patient in group of primary ovarian cancer received concurrent BEV therapy with or without maintenance BEV. The other 42 patients in the group of recurrence received CRS + HIPEC followed by second-line BEV therapy with or without maintenance BEV or second-line BEV therapy with or without maintenance BEV.

Patients were classified as the BEV treatment concurrent group, second-line group, and maintenance group. Patients with primary EOC who underwent upfront debulking surgery and then CP + BEV as first-line systemic therapy were classified as the concurrent group. Patients who underwent post upfront debulking surgery and adjuvant chemotherapy with tumor recurrence then received BEV with chemotherapy were classified as the second-line group. The maintenance group was subdivided into two groups: (1) concurrent therapy followed by maintenance BEV therapy in primary ovarian cancer and (2) second-line therapy followed by maintenance BEV therapy post recurrence. Patients in this retrospective study were divided into the BEV SG or the BEV RG. Patients with persistently high levels of CA-125 during BEV therapy or who experienced tumor progression or recurrence (assessed by CT/PET imaging) within six months posttreatment were classified as the BEV-RG. Patients with normal levels of CA-125 and no tumor progression or recurrence (based on imaging) during or within six months of BEV treatment were classified as the BEV SG group. The regimen of chemotherapy with BEV was based on the GOG-218, ICON-7, and GOG-213 trials. Clinical data were obtained from patient charts and electronic medical records.

### 2.2. Tissue Microarray

Paraffin-embedded tissues from 78 Chinese patients were retrospectively retrieved from the Department of Pathology at Tri-Service General Hospital, including primary tumors or their corresponding omental and peritoneal tumors from these patients who underwent primary cytoreductive surgery. Two pathologists specialized in gynecologic oncology to determine histological subtype and to select the most suitable tissue specimens in the tumor central area for immunohistochemical analysis, and at least two tissue cores (2 mm in diameter) were taken from each of the representative tumor samples and placed in tissue microarray for further study.

### 2.3. Immunohistochemistry Staining

The tissue microarray sections were dewaxed in xylene, rehydrated in alcohol, and immersed in 3% hydrogen peroxide for 10 min to suppress the activity of endogenous peroxidase. Antigen retrieval was carried out by heating each section to 100 °C for 30 min in 0.01 M sodium citrate buffer (pH 6.0). The sections were rinsed three times (5 min each wash) in phosphate-buffered saline (PBS) and then incubated for 1 h at room temperature with rabbit anti-AIM2 antibody (polyclonal; 1:500 dilution; ab93015; AbcamBiotech, Cambridge, UK), anti-C3 antibody (monoclonal; 1:1000 dilution; ab200999; AbcamBiotech, Cambridge, UK) and anti-C5 antibody (polyclonal; 1:300 dilution; ab217027; AbcamBiotech, Cambridge, UK) diluted in PBS. The sections were washed three times (5 min each wash) in PBS, followed by incubation with horseradish peroxidase-labeled immunoglobulin (Dako, Carpinteria, CA, USA) for 1 h at room temperature. The sections were washed three times again, and peroxidase activity was visualized using a solution of diaminobenzidine at room temperature. All tissue microarray slides were examined in terms of immunoreactivity and histological appearance and independently scored concurrently by two of the authors, both gynecological pathologists. Immunoreactivity was graded arbitrarily and semiquantitatively based on the intensity and percentage of staining on the tissue microarray slides.

### 2.4. Evaluation of AIM2, C3 and C5 Expression

We used IHC staining and analyzed expression of these putative biomarkers based on the staining area and intensity of color reaction. The intensity of AIM2, C3 and C5 in individual tumor cells was scored as 0 (no staining), 1+ (weak intensity), 2+ (moderate intensity), 3+ (strong intensity) and 4+ (strongest intensity). The percentages of cells with AIM2, C3 and C5 staining at each intensity were also estimated (range, 0–100). For the semiquantitative analysis of AIM2, C3 and C5 production, the absolute AIM2, C3 and C5 scores were calculated by multiplying the estimated percentages of stained cells at each intensity by the corresponding intensity value, which produced immunostaining scores ranging from 0–400. To compare the absolute AIM2, C3 and C5 scores between the BEV SG and RG, the optimal cutoff values of the AIM2, C3 and C5 scores were determined using the mean value of each biomarker. The slides were processed by substituting the primary antibody with nonimmune serum as a negative control.

### 2.5. Statistical Analysis

All values are expressed as the mean ± standard deviation (SD) and as percentage. Normality was checked using the Kolmogorov-Smirnov-Lilliefors test. A Mann-Whitney U test and chi-square test were performed to compare the AIM2 levels in ovarian tumor samples between the BEV-SG and RG. Associations between AIM2 levels and clinicopathological characteristics were identified using the chi-square test or Fisher’s exact test. The disease-free survival time was monitored by Cox regression analysis. Kaplan–Meier survival curves were compared using the log-rank test. Two-sided *p*-values < 0.05 were considered significant. All analyses were performed using SPSS for IBM, version 21 (IBM Corp., Armonk, NY, USA).

## 3. Results

### 3.1. Clinicopathological Characteristics in 78 Patients

The association between BEV efficacy and clinicopathological features was analyzed (Table 1). The histological subtype was based on the classification model by Shih and Kurman as serous carcinoma (*n* = 58), endometrioid adenocarcinoma (*n* = 4), clear cell carcinoma (*n* = 7), mucinous adenocarcinoma (*n* = 2), and other adenocarcinomas (*n* = 7). As shown in Table 1, the clinical BEV efficacy was not associated with tumor histology or International Federation of Gynecology and Obstetrics (FIGO) stage (*p* = 0.455 and 0.088). Clinical BEV efficacy showed no statistical association with the patient’s age or body mass index (BMI) (*p* = 0.510 and 0.051). The mean course of BEV therapy had no significant association with BEV efficacy (13.4 and 10.9, *p* = 0.189). Complete cytoreduction surgery for primary ovarian cancer had a significant association with the BEV clinical response (24/32 = 75% vs. 8/32 = 25%, *p* = 0.007); suboptimal debulking surgery and CRS + HIPEC post recurrence did not have an association (*p* = 0.112 and 0.277). BEV therapy as a second-line post recurrence and concurrent followed by maintenance in primary cancer had a significant association with BEV efficacy (SG vs. RG: 13/34 = 38.2% vs. 21/34 = 61.8%, *p* = 0.016 and 10/11 = 90.9% vs. 1/11 = 9.1%, *p* = 0.019); concurrent therapy without maintenance therapy in primary cancer and maintenance therapy followed by second-line therapy post recurrence did not show an association with BEV efficacy (*p* = 0.726 and 0.724). The tumor marker CA-125 was partitioned into three levels, 35 U/mL, 70 U/mL, and 105 U/mL, and CA-125 was not associated with the BEV response at these three levels (*p* = 0.499).

The second-line group with or without BEV maintenance therapy (*n* = 42) was also analyzed independently (Table 2). The patients in this group all received adjuvant chemotherapy with paclitaxel and carboplatin for six courses without BEV therapy post debulking surgery. These 42 patients were divided into platinum resistant group (recurrence within six months; *n* = 16) or platinum sensitive group (no recurrence within six months; *n* = 26). The association between platinum sensitive (PS)/resistant (PR) and these three biomarkers (AIM2, C3 and C5) was analyzed. We found that tumor AIM2 level showed no statistical association with clinical platinum sensitive/resistant (PS vs. PR: 313.8 vs. 331.6, *p* = 0.178). There were also no significant differences in C3 (PS vs. PR: 291.3 vs. 304.2, *p* = 0.302) or C5 (PS vs. PR: 221.5 vs. 238.4, *p* = 0.196) regarding platinum sensitive/resistant.

### 3.2. Identification of Predictive Markers

This study’s primary aim was to determine whether any of these biomarkers (AIM2, C3, and C5) were predictive of the clinical advantage of BEV. AIM2, C3 and C5 were expressed mainly in the cytoplasm. AIM2, C3 and C5 expression were also observed in surrounding inflammatory cells, fibroblasts and endothelial cells with heterogenous expression pattern. Although they also have obvious immunostaining in tumor stroma, there were no significant difference of AIM2, C3 and C5 between the BEV SG and BEV RG in these surrounding tumor stroma cells. Therefore, we mainly assessed the immunostaining of tumor cells rather than stroma cells. By evaluating the prevalence of AIM2, C3, and C5 expression in EOC tumors, we found that AIM2 expression was significantly different between the BEV SG and RG (*p* < 0.001) (Table 1). Figure 2A shows the different degrees of IHC staining of AIM2 in tumor cells. A semiquantitative analysis of AIM2 IHC staining scores between the BEV RG and SG was performed, and the AIM2 scores were significantly higher in the BEV RG than in the SG (IHC score 355.5 vs. 297.1, *p* < 0.001) (Table 1 and Figure 2B). However, there was no significant difference between C3 (*p* = 0.077) or C5 (*p* = 0.326) regarding BEV efficacy (Table 1). Examples of IHC staining for C3 and C5 in tumor cells are shown in Figure 2C,E. There was no significant association between the BEV RG and SG based on the semiquantitative analysis (C3: IHC score 306.9 vs. 285.4, *p* = 0.077; C5: IHC score 243.6 vs. 229.3, *p* = 0.326) (Figure 2D,F).

### 3.3. Predictive Associations between IHC Scores for AIM2, C3, and C5 in Tumor Cells and Survival Outcome

For illustrative purposes, Kaplan-Meier plots are presented as AIM2, C3, and C5 dichotomized by the mean value of each IHC score. As shown in Figure 3A, in the entire cohort, patients with AIM2^high^ (mean value > 310) who were treated with BEV had shorter PFS than those with AIM2^low^ (median PFS: 13.58 vs. 19.36 months, *p* < 0.001). Kaplan-Meier survival analysis showed no statistically significant association between C3 and C5 with PFS following BEV therapy (*p* = 0.796 and 0.425). Moreover, there was no significant association between AIM2, C3 and C5 with OS following BEV therapy (*p* = 0.104, 0.623, and 0.344) (Figure 3B). Neither C3 nor C5 was predictive of a therapeutic advantage of BEV. We also used Kaplan-Meier analysis to present OS and PFS times following BEV therapy based on FIGO stage (Figure 4). As expected, patients in more advanced stages had poorer OS and PFS than those in early stages (*p* = 0.062 and 0.002).

### 3.4. Univariate and Multivariate Analyses

As shown in Table 3, the univariate and multivariate analyses revealed that AIM2^high^ conferred worse PFS with BEV therapy than AIM2^low^ (hazard ratio (HR) = 2.79, 95% confidence interval (CI) = 1.50–5.18, *p* = 0.001); patients who received more courses of BEV treatment had a lower risk of recurrence (HR = 0.96, 95% CI = 0.93–0.99, *p* = 0.021); advanced-stage EOC (III, IV) was associated with a higher risk of recurrence than early-stage EOC (I, II) following BEV therapy (HR = 9.41, 95% CI = 2.25–39.4, *p* = 0.002); suboptimal debulking surgery was associated with a higher risk of recurrence than complete cytoreduction surgery following BEV (HR = 2.69, 95% CI = 1.30–5.56, *p* = 0.008); Throughout BEV therapy was associated with better PFS than concurrent therapy or second-line therapy without maintenance BEV therapy (HR = 0.37, 95% CI = 0.15–0.92, *p* = 0.033).

CRS + HIPEC post recurrence seems to confer worse PFS following BEV treatment than initial complete cytoreduction surgery in primary cancer (HR = 1.88, 95% CI = 0.9–3.9, *p* = 0.091); second-line BEV therapy post recurrence seems be associated with inferior PFS than concurrent therapy in primary cancer (HR = 1.82, 95% CI = 0.96–3.46, *p* = 0.067). Cox proportional hazards regression analysis was adjusted for age, BMI, stage, histological type, tumor marker, surgical and BEV therapeutic method. The results revealed an independent effect of AIM2 on PFS with BEV therapy, with higher AIM2 levels being associated with a higher risk of recurrence (adjusted HR = 4.44, 95% CI = 2.01–9.80, *p* < 0.001).

## 4. Discussion

Our study indicates that EOC patients with high expression of tumor AIM2 have a worse response to BEV therapy, accompanied by shorter PFS than those with low expression of tumor AIM2. The level of tissue AIM2 may be a useful molecular biomarker to predict the benefit of BEV therapy in EOC patients. To the best of our knowledge, this is the first study to examine the IHC expression of these tumor tissue immune-related proteins (C3, C5, and AIM2) in EOC patients receiving antiangiogenic therapy. The past decade has uncovered fundamental molecular pathways linking chronic inflammation and cancer [23,24]. In addition to genetic and epigenetic modifications that may evoke unrestrained proliferation and death resistance in cancer cells, the tumor microenvironment is now acknowledged as a critical promoter of tumorigenesis by triggering and supporting local inflammatory processes, angiogenesis, and metastasis [25].

Inflammasome activation leads to chronic inflammation playing a important role in all stages of tumorigenesis such as immunosuppression, proliferation, angiogenesis, and metastasis [17]. Based on our previous study and the result of this research, we propose a working model of a possible link between AIM2 inflammasome and anti-angiogenesis treatment in EOC. In the microenvironment favorable for EOC such as endometriosis or PID, inflammasome is driven directly by specific damage-associated molecular patterns (DAMPs). Extracellular Toll-like receptors (TLRs) identify the specific DAMPs which in turn promote the transcription of pro-inflammatory cytokines or some NLRs (e.g., NLRP3). NLRs and AIM2 assemble into the inflammasome complex which via the caspase recruitment domain (CARD domain) can enroll pro-caspase and facilitate its autocatalytic cleavage to active caspase. Active caspase can cause cell pyroptosis with the effect of the release of inflammatory cytokines and activate pro-inflammatory cytokines that strengthen the inflammatory response. Inflammatory cytokines activate oncogene over-expression then induce EOC carcinogenesis and tumor progression [15]. The inflammatory reaction also has association with the activation of immune cells and recruitment of platelets and circulating leukocytes, all of which can secrete pro-angiogenic factors (VEGF, PIGF and cytokines…) and promote angiogenesis. Neovascularization in tumor cells constitute the permeable immature tumor vessels with lack of vascularization and hyperpermeability resulting in an environment of hypoxia. This resulting hypoxia can contribute to inflammation and may induce the activation of inflammasome related genes (AIM2, NLRP3…) and production of pro-angiogenic factors [26]. Bevacizumab, a humanized monoclonal antibody binds selectively to VEGF-A, thus inhibiting VEGF-A from binding to the VEGFR (tyrosine kinase receptor) to suppress tumor angiogenesis [27] (Figure 5).

In this retrospective study, the patients’ BMI values were not predictive of BEV efficacy, consistent with a previous study based on the GOG-218 trial, in which adiposity was not a predictor for a BEV advantage [28]. ICON-7 trial illustrated that the high-risk population, which included all patients with stage IV and those with unoperated or stage III diseases with suboptimal debulking surgery (residual tumor > 1 cm), derived a better outcome in OS and PFS with BEV addition than chemotherapy alone [6]. However, all patients had received the BEV therapy in our study. Our primary study design was to find clinical or molecular biomarkers to predict the BEV efficacy in these 78 patients with BEV therapy. We have found that BEV efficacy was associated to complete cytoreduction surgery. Suboptimal debulking surgery followed by BEV therapy was associated with a higher risk of recurrence than complete cytoreduction surgery followed by BEV therapy (HR = 2.69, 95% CI = 1.30–5.56, *p* = 0.008). To the best of our knowledge, patients with ovarian cancer receiving complete cytoreduction surgery will have better clinical outcome with subsequent treatment. In the past, research on the role of BEV post CRS + HIPEC was absent. In this study, we found that CRS + HIPEC post recurrence did not show a BEV advantage over primary ovarian cancer post complete cytoreduction surgery. However, this result need further prospective studies to confirm. Patients with primary ovarian cancer who received concurrent BEV therapy followed by maintenance BEV therapy had improved clinical efficacy and a prolonged PFS time, and this result was identical to that of the GOG-218 trial. In this study, the earlier administration and more courses of BEV used had a better therapeutic advantage. Moreover, we found that AIM2 was diversely expressed between the BEV SG and RG and was associated with different PFS outcomes but not OS outcomes. The GOG-218 trial indicated that the improvement in PFS was 3.8 months with maintenance BEV, and in our study, which utilized AIM2, revealed prolonged PFS (of approximately 5.78 months) (median PFS: 19.36 vs. 13.58 months, *p* < 0.001) among patients receiving BEV therapy. This result is consistent with previous theories that the tumorigenesis of ovarian carcinoma may be caused by the inflammatory mechanism, as inflammation is interconnected with angiogenesis [15,29]. The definition of BEV resistant group in our analysis fulfilled the criteria for platinum resistant EOC. In order to know whether the AIM2 level is also platinum sensitive/resistant predictor or not, we have analyzed the second-line group with or without BEV maintenance therapy (*n* = 42). We found that tumor AIM2, C3 and C5 level showed no statistical association with clinical platinum sensitive/resistant.

Currently BEV is recommended for the EOC patients with incomplete cytoreduction surgery at stage III, patients at stage IV or recurrence without confirmed BRCA mutations [6,10]. However, a research based on ICON-7 trial analyzed the PFS and OS within stage IIIB–IV subgroup. All patients in this subgroup (including high risk, all stage IIIB-IV, stage IIIB-IV no residual tumor and stage IIIB-IV with residual tumor), irrespective of residual disease status, derived a PFS benefit from the addition of bevacizumab to chemotherapy. The PFS hazard ratio was 0.77 (95% CI, 0.59–0.99) in 411 patients with stage IIIB–IV ovarian cancer and no visible residuum. In contrast, no OS difference was observed in the subgroup with or without residual disease [30]. Based on the potential benefit for PFS, the CP + BEV was used as adjuvant therapy for 12 patients with primary stage III ovarian cancer post complete cytoreduction in our study.

Our study’s limitations include its retrospective nature and the limited number of patients included, especially those in the early stage, which may have influenced the statistical measurements. The patients enrolled in this study were all ethnic Taiwanese, whereas patients in previous studies were mainly non-ethnic Chinese. The long-term storage stability of tumor tissues may have interfered with the results of IHC staining even though all the samples were treated uniformly. However, two pathologists screened the histological sections and selected areas of representative tumor cells, and two pathologists estimated the IHC staining score. Furthermore, the statistical analysis and pathological assays were performed by research personnel who were blinded to the clinical data.

The search for molecular biomarkers of angiogenesis and anti-angiogenesis and their successful application in the development of antiangiogenic therapy for EOC is a continuous challenge [31]. Until now, there have been no confirmed and accessible biomarkers for routine clinical use to direct patient selection for antiangiogenic therapy [32]. Therefore, molecular and clinical biomarkers are needed to identify patients who are most likely to benefit from antiangiogenic therapies and minimize needless toxicity and medical costs in this “Precision Medicine” era [33].

## 5. Conclusions

Our current study illustrates that the AIM2 inflammasome might be a significant tissue molecular predictor of BEV efficacy. The immunostaining scores of AIM2 were significantly higher in the BEV-RG than in the SG and were associated with PFS. A low level of AIM2 indicated prolonged PFS with BEV therapy (of approximately 5.78 months) (19.36 vs. 13.58 months, *p* < 0.001). However, the molecular mechanism of AIM2 in the tumorigenesis of ovarian cancer and the role of antiangiogenic therapy require further investigation. Ongoing studies will focus on validating the tissue biomarker AIM2 to identify EOC patients who may benefit the most from antiangiogenic therapy.

## Figures and Tables

**Figure 1 jcm-10-04529-f001:**
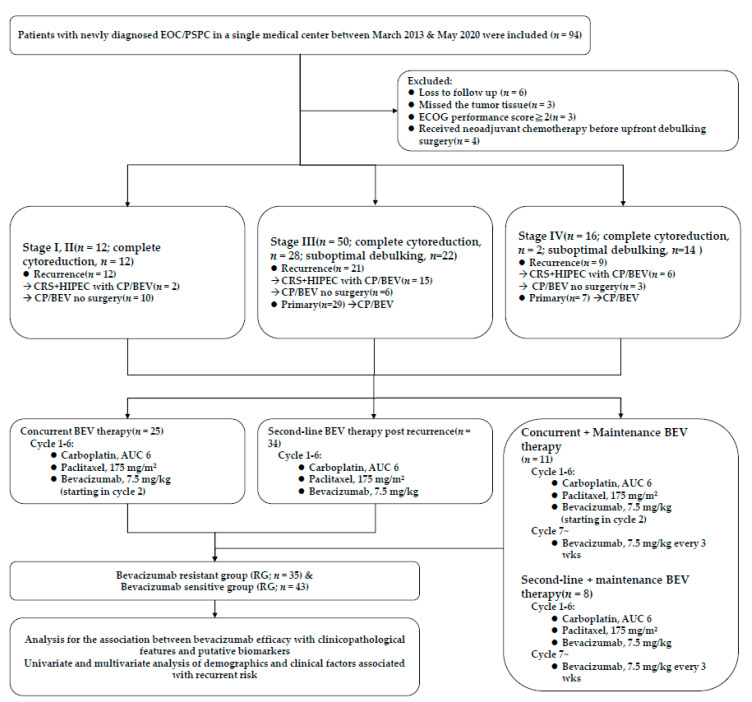
Flow chart of the final patient enrollment. EOC, epithelial ovarian cancer; PSPC, peritoneal serous papillary carcinoma; ECOG, Eastern Cooperative Oncology Group; CRS + HIPEC, cytoreductive surgery and hyperthermic intraperitoneal chemotherapy; CP, carboplatin/paclitaxel; BEV, bevacizumab; AUC, area under curve.

**Figure 2 jcm-10-04529-f002:**
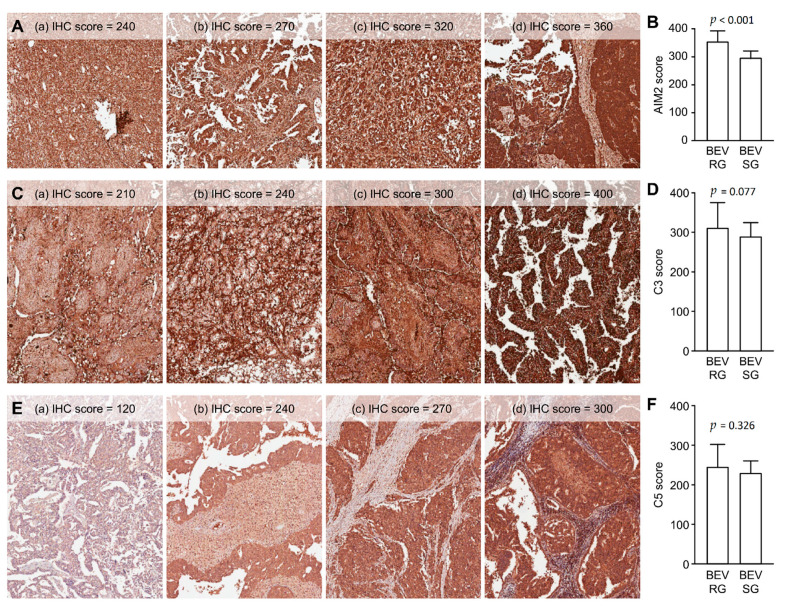
(**A**) Examples of immunohistochemistry (IHC) staining for AIM2 in tumor cells. (**a**) IHC score = +3 × 80%; (**b**) IHC score = +3 × 90%; (**c**) IHC score = +4 × 80%; (**d**) IHC score = +4 × 90%. (**B**) Semiquantitative comparison of AIM2 immunostaining scores between the bevacizumab-resistant group (BEV RG) and bevacizumab-sensitive group (SG). (**C**) Examples of IHC staining for C3 in tumor cells. (**a**) IHC score = +3 × 70%; (**b**) IHC score = +3 × 80%; (**c**) IHC score = +3 × 100%; (**d**) IHC score = +4 × 100%. (**D**) Semiquantitative comparison of C3 immunostaining scores between the BEV RG and SG. (**E**) Examples of IHC staining for C5 in tumor cells. (**a**) IHC score = +2 × 60%; (**b**) IHC score = +3 × 80%; (**c**) IHC score = +3 × 90%; (**d**) IHC score = +3 × 100%. (**F**) Semiquantitative comparison of C5 immunostaining scores between the BEV RG and SG. Absolute IHC score = intensity multiplied by percentages of stained cells.

**Figure 3 jcm-10-04529-f003:**
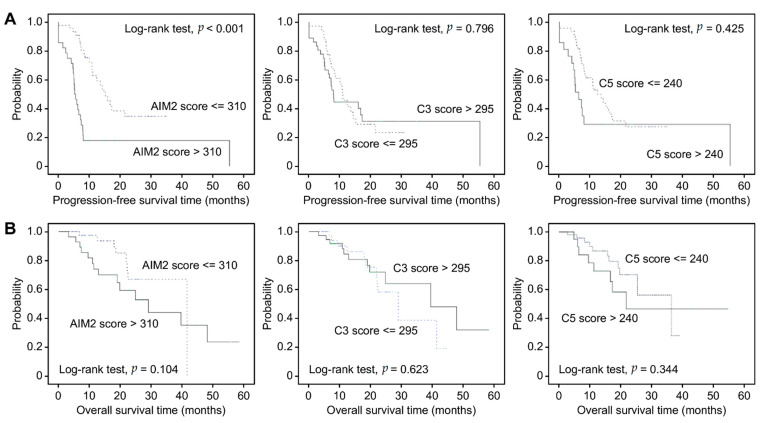
(**A**) Kaplan-Meier analysis of progression-free survival time in EOC patients receiving BEV therapy based on AIM2, C3 and C5 IHC staining scores. (**B**) Kaplan-Meier analysis of overall survival time in EOC patients receiving BEV therapy based on AIM2, C3 and C5 IHC staining scores.

**Figure 4 jcm-10-04529-f004:**
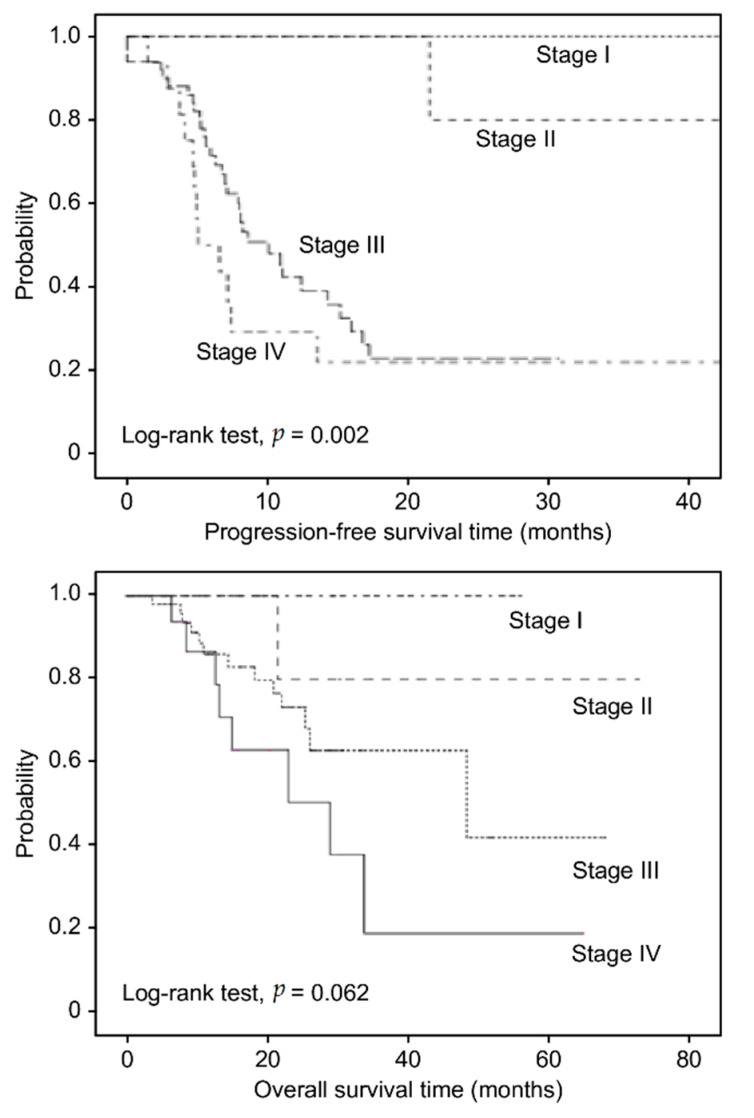
Kaplan-Meier analysis of overall survival and progression-free survival time in EOC patients receiving BEV therapy based on FIGO stage.

**Figure 5 jcm-10-04529-f005:**
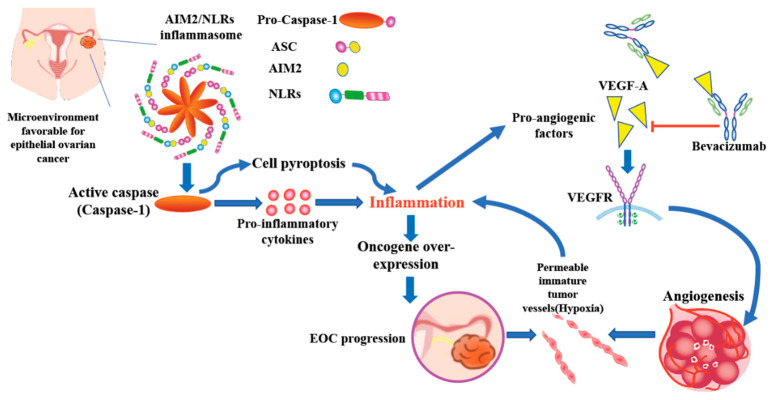
Working model of a possible link between AIM2 inflammasome and anti-angiogenesis treatment in EOC. Hypoxia in ovarian cancer environment induced chronic inflammation and damage associated molecular patterns (DAMPs). Inflammasome related genes (AIM2, NLRP3…) were activated afterward. Inflammatory cytokines which were released due to cell pyroptosis by activated caspase, inducing oncogene over-expression then EOC progression. The inflammatory reaction can also induce production of pro-angiogenic factors and promote angiogenesis. Bevacizumab acts as a direct VEGF inhibitor to suppress tumor angiogenesis.

**Table 1 jcm-10-04529-t001:** Association between bevacizumab efficacy and clinicopathological features.

	Bevacizumab Sensitive Group (SG; *n* = 43)	Bevacizumab Resistant Group (RG; *n* = 35)	
	No. (%)	No. (%)	*p* Value
Age, M(SD)	58.2(9.6)	60.1(8.6)	0.510
BMI, M(SD)	22.8(3.7)	24.5(4.1)	0.051
Number of BEV used times, M(SD)	13.4(9.1)	10.9(7.6)	0.189
Origin			0.456
Ovary	40(57.1)	30(42.9)	
PSPC	3(37.5)	5(62.5)	
Tumor marker			0.499
CA-125 > 35 U/mL	41(53.9)	35(46.1)	
CA-125 > 70 U/mL	40(53.3)	35(46.7)	
CA-125 > 105 U/mL	38(52.1)	35(47.9)	
FIGO stage			0.088
I	6(85.7)	1(14.3)	
II	3(60)	2(40)	
III	29(58)	21(42)	
IV	5(31.2)	11(68.8)	
Histology			0.455
other adenocarcinoma	4(57.1)	3(42.9)	
serous	30(51.7)	28(48.3)	
endometrioid	4 (100)	0(0)	
clear cell	4(57.1)	3(42.9)	
mucinous	1(50)	1(50)	
Surgery			
Complete cytoreduction (CC 0/1)	24(75)	8(25)	0.007
Suboptimal	9(39.1)	14(60.9)	0.112
CRS + HIPEC	10(43.5)	13(56.5)	0.277
BEV therapy			
Maintenance			
Concurrent (Maintenance	10(90.9)	1(9.1)	0.019
Second-line(Maintenance	5(62.5)	3(37.5)	0.724
Concurrent	15(60)	10(40)	0.726
Second-line	13(38.2)	21(61.8)	0.016
AIM2 score, M(SD)	297.1(27.7)	355.5(43.3)	<0.001
≤310	38(88.4)	9(25.7)	<0.001
>310	5(11.6)	26(74.3)	
C3 score, M(SD)	285.4(36.5)	306.9(65.7)	0.077
C5 score, M(SD)	229.3(31.6)	243.6(57.2)	0.326

M(SD), Mean (standard deviation); BMI, body mass index; PSPC, peritoneal serous papillary carcinoma; FIGO, International Federation of Gynecology and Obstetrics; CRS + HIPEC, cytoreductive surgery and hyperthermic intraperitoneal chemotherapy; BEV, bevacizumab; AIM2, absent in melanoma 2; C3, complement component 3; C5, complement component 5.

**Table 2 jcm-10-04529-t002:** Association between platinum sensitive/resistant and putative biomarkers.

	Platinum Sensitive Group (PS; *n* = 26)	Platinum Resistant Group (PR; *n* = 16)	
	No. (%)	No. (%)	*p* Value
AIM2 score, M(SD)	313.8(25.2)	331.6(38.9)	0.178
C3 score, M(SD)	291.3(38.2)	304.2(62.2)	0.302
C5 score, M(SD)	221.5(28.9)	238.4(48.6)	0.196

M(SD) = Mean (standard deviation).

**Table 3 jcm-10-04529-t003:** Univariate and multivariate analyses of demographic and clinical factors associated with recurrence.

	Univariate Analysis		Multivariate Analysis	
	Crude HR (95% CI)	*p* Value	Adjusted HR (95% CI)	*p* Value
Age	1.01 (0.98–1.04)	0.609	1.00 (0.97–1.04)	0.993
BMI	1.06 (0.99–1.13)	0.100	0.96 (0.89–1.04)	0.350
Number of BEV used times	0.96 (0.93–0.99)	0.021	0.98 (0.91–1.06)	0.590
FIGO (III + IV vs. I + II)	9.41 (2.25–39.4)	0.002	26.14 (4.06–168.24)	0.001
Histology (others vs. serous)	0.67 (0.33–1.34)	0.255	0.93 (0.38–2.26)	0.871
CA-125 Surgery	21.33 (0.02–21.27)	0.384	N/A	
Complete cytoreduction (CC 0/1)	1.00 (reference)	(reference)	1.00 (reference)	(reference)
Suboptimal	2.69 (1.30–5.56)	0.008	1.27 (0.55–2.91)	0.578
CRS + HIPEC BEV therapy	1.88 (0.90–3.90)	0.091	1.29 (0.57–2.91)	0.545
Concurrent	1.00 (reference)	(reference)	1.00 (reference)	(reference)
Second-line	1.82 (0.96–3.46)	0.067	1.58 (0.66–3.79)	0.303
Maintenance	0.37 (0.15–0.92)	0.033	0.11 (0.02–0.62)	0.012
AIM2 score (>310 vs. <=310)	2.79 (1.50–5.18)	0.001	4.44 (2.01–9.80)	<0.001

HR, hazard ratio; CI, confidence interval; N/A, Not applicable.

## Data Availability

Not applicable for this study.

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
