# Peer review of "AIM2 Inflammasome in Tumor Cells as a Biomarker for Predicting the Treatment Response to Antiangiogenic Therapy in Epithelial Ovarian Cancer Patients"

_jcm, 2021, doi:10.3390/jcm10194529_

Round 1

Reviewer 1 Report

I do not have any further comments.Minor spell check required.

Author Response

Point:Minor spell check required.

Response :Thanks for reviewer’s helpful recommendation. We will recheck our manuscript and revise the grammatical and spelling errors carefully.

Round 2

Reviewer 1 Report

Immunohistochemical and molecular tests of predictor factors become an important part of the diagnosis of cancer patients. World epidemiological reports indicate a steady increase in the incidence of ovarian cancer. Every year over 114,000 women die from this in the world. Most ovarian cancers are sporadic, and only about 10% of them are genetically determined. Ovarian cancer is the most difficult challenge in oncological gynecology. The prognosis of this disease largely depends on the stage at diagnosis. However, the disease is diagnosed in advanced stage according to FIGO III/ IV. Despite significant advances in understanding  the biology of cancer, debulking surgery and platinum-based chemotherapy is still the gold standard of treatment.
New drugs that would improve treatment outcomes are constantly being searched for. The neoangiogenesis within a cancerous tumor is a hallmark of ovarian cancer. A commonly used drug aimed at inhibiting angiogenesis is bevacizumab. We still do not have any predictive factors that would indicate a group of potential patients who would benefit from such therapy. In particular, we have neither clinical nor molecular biomarkers predictive of response to anti-angiogenic therapy. Therefore, submitted for review study is very interesting as it may contribute to finding a predictive factor.
The authors decided to retrospectively evaluate the treatment results of patients from their center. 78 patients performing primary debulking  surgery were assessed during the 7-year follow-up.  The patients were at different stages of disease, although mainly women in FIGO stage III and IV were included.
Expression of C3, C5 and AIM2 biomarkers was assessed by immunohistochemical staining.  The authors found that AIM2 may be an important predictor of efficacy for BEV therapy.
Low AIM2 expression in tumour cells indicated a significant prolonged median PFS with BEV therapy  (19.36 versus 13.58 months, p <0.001).

The study points to a valuable direction of research and a well-planned retrospective study. However, the molecular mechanism of AIM2 in the oncogenesis of ovarian cancer and its role of anti-angiogenic therapy require further research. Interesting  results of the study obtained by authors need to be confirmed in a larger population of ovarian cancer patients. Further prospective studies are required to confirm these relationships, as well as the molecular mechanism  of this phenomena needs  explanation.
I present the manuscript for further evaluation without comments.

Author Response

Thanks for reviewer’s insightful comments. We reply as below.

Dear reviewer, we are grateful for your insightful comments. The molecular mechanism of AIM2 in the tumorigenesis of ovarian cancer and the role of antiangiogenic therapy require further investigation. Our ongoing studies will focus on validating the tissue biomarker AIM2 to identify EOC patients who may benefit the most from antiangiogenic therapy.

Reviewer 2 Report

This is a well written manuscript investigating an interesting topic that got a bit lost, when Parp-Inhibitor therapy was introduced into the treatment of ovarian cancer. However, there are some issues that have to be clarified.

Major

The effect of multiple markers (AIM2, C3, C5) was assessed in this trial. Therefore, appropriate methods have to be applied to correct for multiple testing.

The group that is defined as BEV resistant in this analysis actually fulfills the criteria for platinum resistant carcinoma. Maybe high AIM 2 levels are an indicator of platinum resistance, rather than BEV resistance? This has to be discussed.

The data of the current anaylsis suggest, that BEV efficacy was associated to optimal tumor debulking. This is conflicting with the results of the ICON 7 trial. You should discuss this results against the background that BEV treatment showed best results in patients with residual tumor in the ICON 7 trial.

As it is probably one of the most important prognostic factors in high grade ovarian cancer data on BRCA mutation status of these patients would be of utmost interest. Do you have any data on the BRCA mutation status of these patients & did the BRCA status have an impact on AIM2 levels?

The discussion is wordy and unfocused.  This section provides extensive information about other factors assessed in ovarian cancer as well as AIM2 levels assessed in other cancers. Thereby, the conclusions of the current work are lost. I would recommend to shorten the discussion significantly. Maybe some of the information can be used in the introduction if deemed absolutely necessary.

Are there any other studies underlining the suggested model of a possible link between AIM 2 inflammasome and anti-angiogenetic treatment? If so, please include in the discussion section.

Minor:

How was optimal debulking surgery and suboptimal debulking surgery defined?

Abstract: „This study was performed to validate biomarkers for prediction of the BEV treatment response in EOC patients“. Check whether validate is the correct word.

Few grammatical and spelling errors have to be corrected e.g. in the discussion section:

„In the breast, it has been shown that AIM2 expression suppresses the proliferation of human breast cancer ells and that AIM2 gene therapy suppresses mammary tumor growth in an orthotopic tumor model [31].“ and „Base on our previous study and the result of this research“

Author Response

Response: Thanks for reviewer’s insightful comments. We reply as below.

Dear reviewer, we are grateful for your insightful comments.  We have made some changes and wish to address these points in detail. Our responses to the reviewer’s comments are as below. We hope that these responses address the reviewer’s concerns. Changes made in the revised text are marked in red.

Major:

Point 1: The effect of multiple markers (AIM2, C3, C5) was assessed in this trial. Therefore, appropriate methods have to be applied to correct for multiple testing.

Response 1: Thanks for reviewer’s insightful comments. All 78 patients in the study had the complete medical records and well-preserved primary tumor tissues. There were two pathologists specialized in gynecologic oncology that screened the histological sections, selected areas of representative tumor cells and estimated the IHC staining score. At least two tissue cores (2 mm in diameter) were taken from each of the representative tumor samples and placed in tissue microarray for further study. We used IHC staining and analyzed expression of these putative biomarkers based on the staining area and intensity of color reaction. In statistical analysis, normality was checked using the Kolmogorov-Smirnov-Lilliefors test. A Mann-Whitney U test and chi-square test were performed to compare the AIM2, C3 and C5 levels in ovarian tumor samples between the BEV-SG and RG. Associations between AIM2 levels and clinicopathological characteristics were identified using the chi-square test or Fisher’s exact test. The disease-free survival time was monitored by Cox regression analysis. Kaplan–Meier survival curves were compared using the log-rank test. Moreover, the statistical analysis and pathological assays were performed by research personnel who were blinded to the clinical data.

Point 2: The group that is defined as BEV resistant in this analysis actually fulfills the criteria for platinum resistant carcinoma. Maybe high AIM 2 levels are an indicator of platinum resistance, rather than BEV resistance? This has to be discussed.

Response 2: Thanks for reviewer’s insightful comments. For this kind recommendation, we have analyzed the second line group with or without BEV maintenance therapy (n=42). The   patients in this group all received adjuvant chemotherapy with Paclitaxel and Carboplatin for six courses without BEV therapy post debulking surgery. These 42 patients were divided into   platinum resistant group (recurrence within six months; n=16) or platinum sensitive group (no recurrence within six months; n=26). The association between platinum sensitive (PS)/resistant (PR) and these three biomarkers (AIM2, C3 and C5) was analyzed. We found that tumor AIM2 level showed no statistical association with clinical platinum sensitive/resistant (PS vs. PR :313.8 vs. 331.6, p=0.178). There were also no significant differences in C3 (PS vs. PR:291.3 vs. 304.2, p=0.302) or C5 (PS vs. PR:221.5 vs. 238.4, p=0.196) regarding platinum sensitive/resistant. We have made explanations in the revised manuscript (page 12, section 4).    

Point 3: The data of the current anaylsis suggest, that BEV efficacy was associated to optimal tumor debulking. This is conflicting with the results of the ICON 7 trial. You should discuss this results against the background that BEV treatment showed best results in patients with residual tumor in the ICON 7 trial.

Response 3: Thanks for reviewer’s helpful recommendation. ICON-7 trial demonstrated that the specific predefined high-risk population, which included all patients with stage IV and those with unoperated or stage III diseases with suboptimal debulking surgery (residual tumor >1 cm), derived a better outcome in OS and PFS with BEV addition than chemotherapy alone: restricted mean OS time was 39.3 months for the BEV group and 34.5 months for the chemotherapy group (log-rank p=0.03); restricted mean PFS time was 20.0 months for the BEV group and 15.9 months for the chemotherapy group (log-rank p=0.001). However, all patients had received the BEV therapy in our study. Our primary study design was to find clinical or molecular biomarkers to predict the BEV efficacy in these 78 patients with BEV therapy. We have found that BEV efficacy was associated to optimal tumor debulking surgery. Suboptimal debulking surgery following BEV therapy was associated with a higher risk of recurrence than optimal debulking surgery following BEV therapy (HR=2.69, 95% CI= 1.30-5.56, p=0.008). To the best of our knowledge, patients with ovarian cancer receiving optimal debulking surgery will have better clinical outcome with subsequent treatment. We have made explanations in the revised manuscript (page 11, section 4). 

Point 4: As it is probably one of the most important prognostic factors in high grade ovarian cancer data on BRCA mutation status of these patients would be of utmost interest. Do you have any data on the BRCA mutation status of these patients & did the BRCA status have an impact on AIM2 levels?

Response 4: Thanks for reviewer’s helpful recommendation. According to recent population-based studies, mutations of the genes BRCA1 and BRCA2 lead to increased cancer predisposition and are present in approximately 14% of epithelial ovarian cancers. The PARP inhibitor was used for the maintenance treatment of patients with platinum-sensitive relapsed BRCA-mutated high-grade serous epithelial ovarian cancer who are in complete or partial response to platinum-based chemotherapy. In our research, there were few patients with high-grade serous epithelial ovarian cancer who received maintenance Olaparib therapy as monotherapy due to BRCA mutations. However, these patients were excluded from this study in the beginning due to the possible influences on progression-free survival and overall survival time. For this kind recommendation, we will collect the data of BRCA mutation status in patients and investigate the relation between BRCA mutation status and some molecular biomarkers.

Point 5: The discussion is wordy and unfocused.  This section provides extensive information about other factors assessed in ovarian cancer as well as AIM2 levels assessed in other cancers. Thereby, the conclusions of the current work are lost. I would recommend to shorten the discussion significantly. Maybe some of the information can be used in the introduction if deemed absolutely necessary.

Response 5: Thanks for reviewer’s helpful recommendation. For this kind recommendation, we will simplify the section of discussion and focus on the connection of AIM2 with ovarian cancer and anti-angiogenetic therapy (page 10, section 4).

Point 6: Are there any other studies underlining the suggested model of a possible link between AIM 2 inflammasome and anti-angiogenetic treatment? If so, please include in the discussion section.

Response 6: Thanks for reviewer’s helpful recommendation. We have searched the primary database such as PubMed and Medline for the model of association between AIM 2 inflammasome and anti-angiogenetic therapy but lack of related articles. Therefore, we propose a working model of a possible link between AIM2 inflammasome and anti-angiogenesis treatment in EOC in the discussion section based on our previous study and the result of this research. However, AIM2 in the tumorigenesis of ovarian cancer and the role of antiangiogenic therapy require further investigation.

Minor:

Point 1: How was optimal debulking surgery and suboptimal debulking surgery defined?

Response 1: Thanks for reviewer’s helpful recommendation. The optimal debulking was defined as a residual tumor that measuring ≤1 cm in diameter after cytoreductive surgery in this study in accordance with Gynecologic Oncology Group (GOG). For this kind recommendation, we have made changes in the revised manuscript (page 5, section 3.1).

Point 2: Abstract: „This study was performed to validate biomarkers for prediction of the BEV treatment response in EOC patients“. Check whether validate is the correct word.

Response 2: Thanks for reviewer’s helpful recommendation. For this kind recommendation, we have changed the word “validate” to “search” to make the abstract smoother. We have made changes in the revised manuscript (page 1, abstract).

Point 3: Few grammatical and spelling errors have to be corrected e.g. in the discussion section:

„In the breast, it has been shown that AIM2 expression suppresses the proliferation of human breast cancer ells and that AIM2 gene therapy suppresses mammary tumor growth in an orthotopic tumor model [31].“ and „Base on our previous study and the result of this research“

Response 3: Thanks for reviewer’s helpful recommendation. We will recheck our manuscript and revise the grammatical and spelling errors carefully.
